# SHAKE THE $k$-CENTER: TOWARD NOISE-ROBUST CORESETS VIA RELIABILITY SWAPPING BETWEEN NEIGHBORS

## ABSTRACT

Coreset selection aims to select a small high-quality subset from a large-scale dataset to support DNN downstream tasks. $k$-Center is a solid coreset selection approach with a theoretical guarantee. It considers coresets from the covering theory: In the feature space, a coreset can cover all data with a sphere centered on each sample in the coreset. Smaller covering radii indicate better quality. However, the performance of $k$-center degrades and lags behind other methods on noisy datasets. To the best of our knowledge, there is still a lack of explanation for this phenomenon. We propose a theory for this phenomenon. Our theory indicates that the noise rate of the coreset constrains the generalization performance of the selected subset. With this theory, we propose a coreset selection method under label noise, named Shaker, whose core idea is to jointly optimize the set cover and reliability of the coreset. Shaker first generates a batch of candidates with a small covering radius and then swaps in their reliable neighbors while maintaining a good set cover. Extensive results demonstrate that Shaker outperforms baseline methods by up to 14.3%.

## 1 INTRODUCTION

Coreset selection is an important supporting technology for DNN models. It reduces the original large-scale dataset to a small high-quality subset (Guo et al., 2022) to serve downstream tasks such as accelerating neural architecture search (C et al., 2022), constructing memory for continuous learning (Yoon et al., 2022), and active learning (Killamsetty et al., 2021c). For example, neural architecture search and hyperparameter fine-tuning are key steps towards high-performance models. They require training hundreds to thousands of models using the same dataset, showing a major development bottleneck. Coreset alleviates this issue by reducing the training iterations with a high-quality subset. Coreset technology speeds up this process by more than 8x while sacrificing accuracy within 1% (C et al., 2022).

$k$-Center is a theoretically guaranteed coreset selection approach (Sener & Savarese, 2018). Its core idea is that in the feature space, a coreset can cover all samples of the full dataset with a sphere centered in each sample of the coreset. Smaller covering radii of the sphere imply better representations for the original full dataset. Optimizing the covering radii of $k$-center demonstrates solid performance across many tasks (Guo et al., 2022).

Constructing a noise-robust coreset is a crucial task. The training labels are often unreliable. For instance, Wei *et al.* relabeled the CIFAR-10 dataset on the Amazon Mechanical Turk platform. The noise rate of the new labels is as high as 18% (Wei et al., 2022)! However, if we place $k$-center under label noise, we observe that the performance of $k$-center degrades and lags behind the compared methods by a large margin. To the best of our knowledge, there is still a gap in explaining the performance degradation of $k$-center under label noise.

We propose a theory to explain this phenomenon. Specifically, we introduce noisy labels in the set cover theory (Sener & Savarese, 2018). It leads to an interesting upper bound, which suggests that the training error of the noisy coresets is upper bounded by the noise loss of the original dataset, the coreset selection approximation loss from the ideal clean dataset, and the noise loss of the selected coreset. Our complete theory demonstrates that the noisy coreset selection loss is dominated by

the covering radius and the coreset noise rate. Empirical experiments support our theory. However, incorporating two factors in coreset selection is challenging due to the potential seesaw effect.

We propose Shaker for robust coreset selection under label noise. Its core idea is similar to shaking $k$-center: it optimizes the noise rate of $k$-center by swapping in reliable neighbors of the $k$-center while maintaining a small covering radius. First, we generate a batch of candidates by optimizing the covering radii in the feature space. Second, we swap in their reliable neighbors with small losses (Han et al., 2018) into the coreset. We formalize this swapping as a bipartite graph matching problem, where one side is the candidates, the other side is the remaining samples (including themselves so that they can also be retained), and the connection edges between both sides are weighted by the neighbors' reliability improvement with the consideration of the distance to its nearest sample center.

We implement Shaker using PyTorch. Under the same software and hardware configurations, we compare Shaker with other baselines on manually relabeled CIFAR-10N, CIFAR-100N, and large-scale WebVision dataset. Extensive experimental results show that Shaker leads to better generalization performance. Improving the efficiency of Shaker is future work.

## 2 BACKGROUND & MOTIVATION

**Problem Statement.** We consider the coreset selection problem under label noise. Suppose a dataset $\mathcal{D} = \{(\mathbf{x}_i, y_i)\}_{i=1}^n$ contains $C$ classes. Coreset selection aims to extract a high-quality subset $\mathcal{S}$ of size $m$ from $\mathcal{D}$ to serve downstream tasks such as neural architecture search acceleration and continuous learning. However, due to the imperfection of annotators in the real world, sample $\mathbf{x}_i$ is labeled as wrong $\widetilde{y}_i$ with a certain noise rate, which degrades the model performance trained with it. How to enable the coreset to shield label noise is crucial.

### 2.1 REVIEW OF $k$-CENTER

$k$-Center selects coreset based on set cover. It views coreset $\mathcal{S}$ to be a $\delta_\mathcal{S}$ cover of the entire dataset $\mathcal{D}$, which means that the balls of radius $\delta_\mathcal{S}$ centered at each sample in $\mathcal{S}$ can cover the entire $\mathcal{D}$.

Previous work demonstrated that the quality of $k$-center depends on decreasing covering radius over the size of the coreset. Denote the trainable model weights as $\mathbf{w}$ and the loss function as $l(\mathbf{x}, y; \mathbf{w})$. If the loss function $l(\mathbf{w}, y; \mathbf{w})$ is $\lambda^l$-Lipschitz continuous and bound by $L$. For each class $c$, its regression function $\eta_c(\mathbf{x}) = p(y = c|\mathbf{x})$ is $\lambda^\eta$-Lipschitz continuous. With the assumptions in (Sener & Savarese, 2018), $k$-center's upper bound holds at least $1 - \gamma$:

$$\left| \frac{1}{n} \sum_{i \in \mathcal{D}} l(\mathbf{x}_i, y_i; \mathbf{w}) - \frac{1}{m} \sum_{j \in \mathcal{S}} l(\mathbf{x}_j, y_j; \mathbf{w}) \right| \leq \delta_\mathcal{S}(\lambda^l + \lambda^\eta LC) + \sqrt{\frac{\log(1/\gamma)L^2}{2n}}. \tag{1}$$

### 2.2 THE PERFORMANCE GAP BETWEEN $k$-CENTER AND SOTA UNDER LABEL NOISE

However, the solid $k$-center can not extend to datasets with noisy labels. Figure 1 reports this phenomenon (abbreviated as P1 in the following paper). Specifically, this task is to select 20% subsets from the datasets and evaluate its test accuracy with a variant of ResNet-18. The experiments include two CIFAR-10 datasets with different noise rates and one noisy CIFAR-100 dataset. The results show that $k$-center lags behind SOTA by 2.2%, 3%, and 8.1%, respectively[1]. It highlights a key limitation of the $k$-center method. To the best of our knowledge, understanding why this phenomenon happens is still a gap.

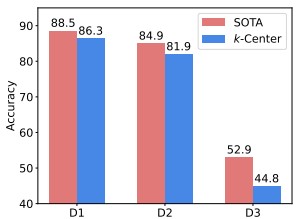

Figure 1: $k$-Center lags behind SOTA on three noisy datasets.

---

[1]The set of data are from (Park et al., 2023)

## 3 WHY DOES $k$-CENTER LAG IN THE PRESENCE OF LABEL NOISE?

### 3.1 THEORETICAL EXPLANATION

We consider the effect of label noise in Eq. 1. By applying the triangle inequality, we decompose the total selection error into:

$$\left| \frac{1}{n} \sum_{i \in \mathcal{D}} l(\mathbf{x}_i, \tilde{y}_i; \mathbf{w}) - \frac{1}{m} \sum_{j \in \mathcal{S}} l(\mathbf{x}_j, \tilde{y}_j; \mathbf{w}) \right| \leq \underbrace{\left| \frac{1}{n} \sum_{i \in \mathcal{D}} l(\mathbf{x}_i, \tilde{y}_i; \mathbf{w}) - \frac{1}{n} \sum_{i \in \mathcal{D}} l(\mathbf{x}_i, y_i; \mathbf{w}) \right|}_{\text{(I) Noisy Full Dataset Loss}} +$$

$$\underbrace{\left| \frac{1}{n} \sum_{i \in \mathcal{D}} l(\mathbf{x}_i, y_i; \mathbf{w}) - \frac{1}{m} \sum_{j \in \mathcal{S}} l(\mathbf{x}_j, y_j; \mathbf{w}) \right|}_{\text{(II) Coreset Selection Loss}} + \underbrace{\left| \frac{1}{m} \sum_{i \in \mathcal{S}} l(\mathbf{x}_i, y_i; \mathbf{w}) - \frac{1}{m} \sum_{j \in \mathcal{S}} l(\mathbf{x}_j, \tilde{y}_j; \mathbf{w}) \right|}_{\text{(III) Noisy Coreset Loss}}. \quad (2)$$

This step outlines that the loss of coreset selection from noisy datasets is bounded by three terms. (I) is controlled by the intrinsic noise of the original datasets. (II) is the loss of clean coreset selection. (III) represents the noise effect in the extracted coresets. With this observation, We state the following theorem:

**Theorem 1.** *Suppose the label noise of selected coresets is bounded by $\epsilon$. Under the same assumptions as in $k$-center (Sener & Savarese, 2018), the following holds at least $1 - \gamma$:*

$$\left| \frac{1}{n} \sum_{i \in \mathcal{D}} l(\mathbf{x}_i, \tilde{y}_i; \mathbf{w}) - \frac{1}{m} \sum_{j \in \mathcal{S}} l(\mathbf{x}_j, \tilde{y}_j; \mathbf{w}) \right| \leq \delta_{\mathcal{S}}(\lambda^l + \lambda^\eta LC) + \epsilon L + \sqrt{\frac{\log(1/\gamma)L^2}{2n}}. \quad (3)$$

*Proof.* Eq. 2 indicates that (I) is a constant in coreset selection, and (II) is the exact Eq. 1. For (III), its upper bound $\leq \frac{L}{m} \sum_{j \in \mathcal{S}} 1_{\tilde{y}_j = y_j} = \epsilon L$. Combining the above results, we can get Eq. 3. □

This theorem suggests that we can bound the coreset loss with covering radius, noise rate, and a term close to zero with increasing $n$. Therefore, one explanation for P1 is that *$k$-center does not take the sample noise into account in coreset selection.*

### 3.2 HOW TO APPLY THEOREM 1 TO CORESET SELECTION? A PRIMARY CHALLENGE

Before applying Theorem 1 to coreset selection, we are interested in two questions. (Q1): Is Theorem 1 supported by empirical evidence? (Q2): What are the challenges in applying Theorem 1 to coreset selection?

To answer both questions, we design a careful experiment. Our design principle is to generate a series of coresets by linearly controlling a factor (covering radius or noise rate). We then check whether the suggestion of Theorem 1 is aligned with the results. The experiment uses CIFAR-10N dataset with a 40% noise rate and a deep model from ResNet-18 family. Considering the effectiveness of the small loss criterion in identifying clean labels, we construct a series of subsets of 10% size in descending order of sample loss. We calculate their noise rate, covering radius, and the corresponding accuracy. Figure 2 shows the effect of the two factors. The larger and darker circles indicate better accuracy.

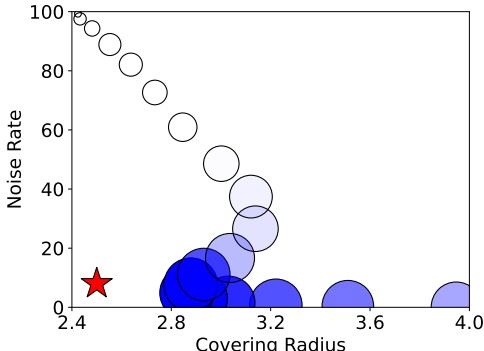

Figure 2: Impact of covering radius and noise rate on test accuracy.

First, the experimental results firmly support Theorem 1. Along the noise rate or the covering radius dimension, increasing values consistently lead to decreased accuracy (Answer to Q1). Second,

moving the coreset toward the lower-left corner of the plot is challenging. Because the covering radius and noise rate are orthogonal, improving one could degrade the other. As shown in Figure 2, although coresets along the reverse direction of main diagonal achieve smaller covering radii, they exhibit higher noise rates, resulting in lower accuracies. Hence, how to shift coresets toward the lower-left corner is challenging (Answer to Q2).

# 4    OUR METHOD: SHAKER

Figure 3: The diagram of Shaker.

**A Bird's Eye View.** We propose Shaker to move towards generating coresets with small covering radius and noise rate under label noise. Its core idea is to optimize the noise rate of $k$-center by swapping out unreliable sample points and swapping in their reliable neighbors based on the small-loss criterion as in Figure 3. To be specific, it consists of the following three steps. First, Shaker identifies $k$-center by reducing the covering radius. Second, Shaker shakes these points through an efficient bipartite graph matching to exchange some sample points and their reliable neighbors so as to reduce the noise rate while maintaining a small covering radius. Finally, Shaker generates the final coreset by making up for the mistakes in the previous step by alternating the above two steps.

## 4.1    LEFT FOOT: OPTIMIZE THE COVERING RADIUS OF THE CORESET

This step aims at generating a batch of coreset candidates by minimizing the covering radii. Formally, for a batch coreset candidates $\mathcal{S}'$ of size $B$, their selection is

$$\min_{|\mathcal{S}'| \leq B} \max_{i \in \tilde{\mathcal{D}} \setminus (\mathcal{S} \cup \mathcal{S}')} \min_{j \in \mathcal{S} \cup \mathcal{S}'} \triangle(e(\mathbf{x}_i), e(\mathbf{x}_j)), \tag{4}$$

where $e(\cdot)$ is the features of samples. Eq. 4 is a minimax facility location problem (Wolf, 2011). Solving it is NP-Hard (Cook et al., 1998). We adopt a naive greedy selection solution following (Sener & Savarese, 2018). Specifically, we first assign a center for all unselected samples with the nearest sample in the current $S$. Then we select the farthest sample to its corresponding center into the batch candidate $\mathcal{S}'$. This process is executed incrementally until $B$ candidates are acquired. It is theoretically guaranteed that the resulting covering radii is upper bound by two times of the optimal radii (Sener & Savarese, 2018).

## 4.2    RIGHT FOOT: REDUCE THE NOISE RATE OF THE CORESET BY SWAPPING IN RELIABLE NEIGHBORS

This step aims to refine the candidates to be more reliable. Our refinement method leverages the concept of bipartite graph matching in graph theory. A bipartite graph is a special type of undirected graph. Its vertex set can be divided into two disjoint subsets such that each edge connects vertices in two vertex sets. The goal of bipartite graph matching is to minimize the sum of the connection weights between two point sets.

We formalize the swapping between the candidates and their reliable neighbors with the bipartite graph matching. Specifically, we treat the coreset candidate $\mathcal{S}'$ as one vertex set. We treat the

remaining unselected samples $\tilde{\mathcal{D}}\backslash\mathcal{S}$ as the other vertex set. Both sets are considered as independent and disjoint so that samples in $\mathcal{S}'$ can connect with themselves and stay in coreset. We design the edge weight $C_{k,i}$ between sample $k$ and $i$ from two sets with the reliability improvement

$$C_{k,i} = -(1 + \exp(-l_i/\tau))^{\exp(-\triangle(e(\mathbf{x}_k), e(\mathbf{x}_i)))}, \tag{5}$$

where $\tau$ is a coefficient that controls the swapping strength. Eq. 5 incorporates two considerations: (1) Closer neighbors are preferred during swapping. In Eq. 5, such neighbors lead to a larger exponent of the exponential formula so that $C_{k,i}$ is smaller (because of the minimization objective below). (2) Neighbors with small losses are preferred during swapping. The small-loss criterion demonstrates that samples with small training loss are more likely to be correctly labeled and reliable. In Eq. 5, such neighbors lead to a larger base of the exponential formula so that $C_{k,i}$ is smaller. The final optimization goal is

$$\min \quad \sum_{k \in \mathcal{S}'} \sum_{i \in \tilde{\mathcal{D}}\backslash\mathcal{S}} C_{k,i} Z_{k,i} \,,$$

$$s.t. \quad \sum_{i \in \tilde{\mathcal{D}}\backslash\mathcal{S}} Z_{k,i} = 1 \,, \quad \forall k \in \mathcal{S}' \,,$$

$$\sum_{k \in \mathcal{S}'} Z_{k,i} \leq 1 \,, \quad \forall i \in \tilde{\mathcal{D}}\backslash\mathcal{S} \,,$$

$$Z_{k,i} \in \{0, 1\} \,, \tag{6}$$

where $Z_{k,i}$ is the decision assignment variable indicating whether $k$ and $i$ are connected and should swap in the $i$-th sample. Eq. 6 can be solved efficiently in polynomial time using the augmenting path algorithm, *i.e.*, LAPJVsp (Jonker & Volgenant, 1987).

### 4.3 Generate a Coreset by Alternating the Two Steps

We repeat the above two steps to generate the next batch of samples until the complete coreset is generated. On the one hand, the right foot may swap out samples with significant impacts on the covering properties, hurting the representativeness of the coreset. Our alternating running could correct the potential swapping errors. On the other hand, It can save computing resource. The size of a full dataset is often tens of thousands, limiting the scalability of coreset selection. Alternating running in batches allows Shaker to be applied to large datasets. Algorithm 1 concludes Shaker. For brevity, we omit the remainder processing in batch splitting, which uses the same code.

---

**Algorithm 1** Shaker algorithm.

---

**Input**: $\forall(\mathbf{x}_i, \tilde{y}_i) \in \tilde{\mathcal{D}}$ and the coreset budget $m$.
**Parameter**: Batch size $B$. Swapping coefficient $\tau$.
**Output**: Coreset $\mathcal{S}$.
1: Let $\mathcal{S} = \varnothing$.
2: **for** batch $1,2,\cdots,m//B$ **do**
3:     Let $\mathcal{S}' = \varnothing$.
4:     **repeat**
5:       **if** $\mathcal{S}$ is $\varnothing$ **then**
6:         Init $\mathcal{S}'$ using the sample with the smallest loss.
7:       **end if**
8:       $k = \arg\max_{i \in \tilde{\mathcal{D}}\backslash(\mathcal{S}\cup\mathcal{S}')} \min_{j \in \mathcal{S}\cup\mathcal{S}'} \triangle(e(\mathbf{x}_i), e(\mathbf{x}_j))$.
9:       $\mathcal{S}' \leftarrow \mathcal{S}' \cup \{k\}$.
10:     **until** $|\mathcal{S}'| = B$.
11:     Compute cost matrix $C$ according to Eq. 5.
12:     Solve Eq. 6 with LAPJVsp algorithm.
13:     Add the solved points into $\mathcal{S}$.
14: **end for**
15: **return** $\mathcal{S}$.

---

## 5 EXPERIMENT

### 5.1 SETTING

**Datasets.** We use three datasets with real-world noisy labels. (1) CIFAR-10N (Wei et al., 2022). It contains 10 classes. Each sample pairs with human-annotated labels collected from Amazon Mechanical Turk. It contains two noisy versions: one with 18% and one with 40%. (2) CIFAR-100N (Wei et al., 2022). It provides 100 classes. Each image has been re-labeled by humans like CIFAR-10N. (3) WebVision. It is built by crawling from the Web using 1,000 concepts. We use Google version by following (Chen et al., 2019; Park et al., 2023).

**Baselines.** We compare the following eight baselines: Uniform, $k$-center (Sener & Savarese, 2018), SmallLoss (Jiang et al., 2018), Margin (Coleman et al., 2020), Forgetting (Toneva et al., 2019), GraNd (Paul et al., 2021), Moderate (Xia et al., 2023b), and Prune4ReL (Park et al., 2023).

**Evaluation Details.** For CIFAR-10N, we train PreAct Resnet-18 (He et al., 2016) for 300 epochs with a batch size of 128. The used optimizer is an SGD with a weight decay of 0.0005. Its initial learning rate is 0.02 and decayed with a cosine annealing scheduler. We use SOP+ (Liu et al., 2022) to enhance the noise-robust training. For CIFAR-100N, we adopt similar settings except that the noise-robust training uses DivideMix (Li et al., 2020). For WebVision, we train InceptionResNetV2 (Szegedy et al., 2017) with an SGD optimizer. Its initial learning rate is 0.02 and dropped by a factor of 10 at the 50th epoch. The training process consists of 100 epochs. The batch size used is 32. We use SL (Wang et al., 2019) as noise-robust training. In coreset selection, we generate coresets based on pre-warmed CIFAR-10N of 10 epochs, CIFAR-100N of 30 epochs, and WebVision (Guo et al., 2022; Park et al., 2023) of 10 epochs. Considering that downstream tasks such as NAS prefer to use small coresets to achieve better acceleration, we primarily evaluate three coreset sizes: 0.05, 0.15, and 0.25. For Shaker, the batch size $B$ is set to 2500. The $\tau$ of CIFAR-10N and CIFAR-100N are configured to 0.1, 0.2, and 0.3 for three coreset sizes. For WebVision, we set $\tau$ with 0.1, 0.5, and 0.9. We run each experiment three times and report the statistics of final accuracy.

### 5.2 END-TO-END COMPARISON

**Performance on Varying Noise Rates.** Table 1 shows the comparison results of test accuracy on CIFAR-10N. Shaker consistently achieves the best performance across both noise levels. To be specific, Shaker outperforms the compared methods by margins of up to 14.3% on CIFAR-10N. Prune4ReL performs well in most tasks but is still weaker than our proposed Shaker. On the CIFAR-10N ($\approx 40\%$) task, Shaker outperforms Prune4ReL by up to 14.3%. Although Uniform achieves high performance in some cases due to its purposeless mediocrity in noise ratio and geometric distribution, its effect diminishes if it is applied to high noise ratio or complex classification tasks. For example, Uniform falls behind Shaker by 15.1% on the CIFAR-10N. The performance of Moderate is also weaker than our method. To some extent, Margin, Forget, and GraNd all favor hard samples since they are fascinated by their rich information, which are unreliable under noisy conditions. This also results in suboptimal performance. For example, the best performance of GraNd on CIFAR-10N ($\approx 40\%$) is 13.7%, 72.7% behind our Shaker. Relying solely on covering radius or training loss is

Table 1: Test accuracy (%) comparisons on CIFAR-10N with SOP+.

| Selection Methods | Noise Rate$\approx$18% | | | Noise Rate$\approx$40% | | |
|---|---|---|---|---|---|---|
| | 0.05 | 0.15 | 0.25 | 0.05 | 0.15 | 0.25 |
| Uniform | 72.4$\pm$0.5 | 85.5$\pm$0.5 | 88.9$\pm$0.2 | 61.9$\pm$0.6 | 78.7$\pm$0.3 | 83.8$\pm$0.2 |
| SmallL | 42.0$\pm$5.5 | 63.2$\pm$1.2 | 76.1$\pm$3.0 | 49.2$\pm$3.0 | 70.4$\pm$2.6 | 80.9$\pm$0.6 |
| Margin | 32.8$\pm$4.5 | 43.4$\pm$5.1 | 54.8$\pm$3.5 | 28.6$\pm$0.8 | 35.8$\pm$2.1 | 44.5$\pm$1.3 |
| kCenter | 59.9$\pm$2.5 | 83.7$\pm$0.6 | 89.1$\pm$0.2 | 47.4$\pm$1.8 | 71.3$\pm$1.2 | 82.1$\pm$0.8 |
| Forget | 55.9$\pm$1.5 | 74.3$\pm$0.9 | 86.4$\pm$0.5 | 53.3$\pm$0.4 | 67.3$\pm$0.4 | 77.4$\pm$0.3 |
| GraNd | 10.4$\pm$0.1 | 17.4$\pm$2.7 | 26.7$\pm$6.4 | 8.5$\pm$4.0 | 11.7$\pm$1.4 | 15.6$\pm$1.5 |
| Moderate | 74.7$\pm$0.6 | 87.2$\pm$0.1 | 90.3$\pm$0.3 | 59.3$\pm$1.6 | 74.5$\pm$0.6 | 77.9$\pm$1.4 |
| Pr4ReL | 75.8$\pm$1.2 | 85.6$\pm$0.9 | 89.2$\pm$0.5 | 62.7$\pm$5.9 | 79.1$\pm$1.9 | 83.8$\pm$0.8 |
| Shaker | **78.5**$\pm$0.2 | **87.5**$\pm$0.6 | **90.7**$\pm$0.1 | **77.0**$\pm$1.2 | **84.5**$\pm$0.5 | **86.9**$\pm$0.1 |

insufficient, particularly when the coreset size is constrained. For instance, $k$-center and SmallLoss lag behind Shaker by 29.6% and 27.8%. This highlights the necessary combination of reducing the noise rates and covering radius.

Table 2: Test accuracy (%) comparisons on CIFAR-100N on DivMix.

| Coreset Size | Selection Methods | | | | | | | | |
|---|---|---|---|---|---|---|---|---|---|
| | Uniform | SmallL | Margin | kCenter | Forget | GraNd | Moderate | Pr4ReL | Shaker |
| 0.05 | 12.6±2.0 | 11.5±1.0 | 6.3±0.3 | 12.8±0.9 | 14.4±7.6 | 6.9±0.8 | 13.3±2.1 | 16.0±0.9 | **16.1**±0.4 |
| 0.15 | 27.1±0.8 | 26.9±2.9 | 13.1±0.5 | 29.3±1.4 | 29.3±1.8 | 12.5±1.0 | 25.1±1.8 | 27.1±4.4 | **34.3**±2.7 |
| 0.25 | 36.7±3.6 | 32.8±2.4 | 19.3±2.1 | 37.8±1.5 | 38.9±2.8 | 15.9±0.4 | 36.1±3.3 | 40.2±4.1 | **40.3**±2.6 |

Table 3: Test accuracy (%) comparisons on WebVision with SL.

| Coreset Size | Selection Methods | | | |
|---|---|---|---|---|
| | SmallL | kCenter | Pr4ReL | Shaker |
| 0.05 | 23.3±0.9 | 30.6±0.9 | 33.5±1.5 | **36.6**±0.6 |
| 0.15 | 37.5±1.0 | 45.7±0.7 | 46.1±0.8 | **48.0**±0.8 |
| 0.25 | 44.5±1.3 | 51.5±1.9 | 52.6±0.4 | **53.0**±0.3 |

**Performance on Varying Datasets and Training Methods.** We further evaluate the transferability of Shaker with CIFAR-100N and WebVision. Table 2 and 3 compares the performance of Shaker with baselines. The results show that Shaker consistently achieves the best performance. In CIFAR-100N, Shaker outperforms baselines by 5%. In WebVision, Shaker outperforms SmallLoss, kCenterGreedy and the Prune4Rel by 3.1%. These results demonstrate that effectively optimizing noise rate and covering radius could lead to more robust performance.

## 5.3 IN-DEPTH ANALYSIS

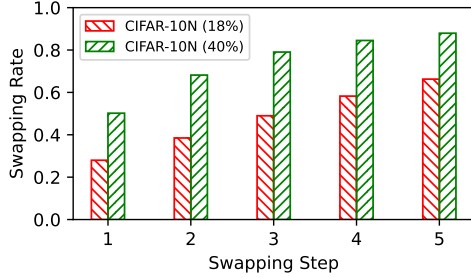

Figure 4: Relationship between swapping step and swapping rate.

**Swapping Rate.** Figure 4 depicts the relationship of the swapping rate of Shaker along the swapping steps. The results are from coresets of 0.25 size on CIFAR-10N. We obtain two significant observations: (1) The higher the noise rate, the higher the swapping rate. This underscores the effect of Shaker on reducing the noise rate of coresets. (2) The swapping rate increases with the number of swapping steps. This may be because the noise rate of the remaining unselected datasets gradually increases during the selection. The neighbor swapping mechanism responsively increases swapping rates to maintain a low noise rate.

**Noise Rate.** Table 4 shows the noise rates of the selected subsets. As we can see, the noise rates of Shaker are lower than that of other methods but higher than that of SmallLoss. This can be attributed to the consideration for covering radius. SmallLoss simply considers the noise rates of coresets. Due to the uneven distribution, the generalization performance is limited. Other coreset methods do not take the noise rate into account. High label noise results in suboptimal generalization performance.

## 5.4 ABLATION STUDIES

**The Effect of Swapping Coefficient.** The swapping coefficient $\tau$ controls the swapping strength between sample loss and covering radius. Large $\tau$ prefers clean samples to optimize the label noise,

Table 4: Noise rates (%) of the selected subsets (CIFAR-10N).

| Selection Methods | Noise Rate≈18% | | | Noise Rate≈40% | | |
|---|---|---|---|---|---|---|
| | 0.05 | 0.15 | 0.25 | 0.05 | 0.15 | 0.25 |
| Uniform | 18.3 ±0.2 | 17.9 ±0.6 | 17.9 ±0.5 | 41.8 ±0.8 | 40.8 ±0.3 | 40.5 ±0.2 |
| SmallL | 0.0 ±0.0 | 0.0 ±0.0 | 0.1 ±0.0 | 0.1 ±0.1 | 0.4 ±0.1 | 1.0 ±0.3 |
| Margin | 32.3 ±1.5 | 31.1 ±1.4 | 29.2 ±1.0 | 56.4 ±0.6 | 56.0 ±0.1 | 54.7 ±0.0 |
| kCenter | 18.9 ±0.6 | 19.2 ±0.5 | 19.3 ±0.4 | 41.9 ±1.1 | 42.9 ±0.6 | 42.7 ±0.5 |
| Forget | 20.1 ±1.0 | 18.0 ±0.1 | 17.3 ±0.1 | 39.2 ±0.3 | 36.6 ±0.4 | 35.8 ±0.2 |
| GraNd | 94.2 ±2.2 | 78.2 ±3.8 | 61.8 ±2.3 | 99.1 ±0.2 | 95.3 ±0.7 | 89.6 ±0.7 |
| Moderate | 6.2 ±0.4 | 6.3 ±0.2 | 6.5 ±0.1 | 31.2 ±0.6 | 31.4 ±0.4 | 31.8 ±0.3 |
| Pr4ReL | 9.9 ±0.1 | 13.3 ±0.2 | 15.3 ±0.2 | 26.3 ±0.7 | 32.7 ±0.4 | 35.7 ±0.6 |
| Shaker | 2.5 ±0.4 | 4.4 ±0.2 | 6.1 ±0.2 | 2.1 ±0.3 | 8.5 ±0.3 | 13.8 ±0.4 |

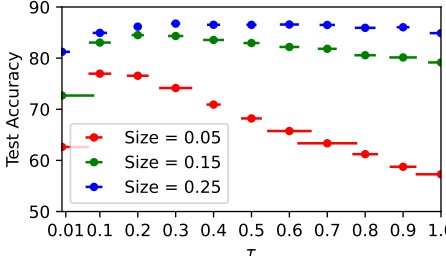

Figure 5: The effect of swapping coefficient. (The horizontal lines indicate error bars.)

Table 5: The impact of batch size.

| Coreset Size | Batch Size | | | |
|---|---|---|---|---|
| | 100 | 250 | 500 | 2500 |
| 0.05 | 76.1±0.3 | 76.0±0.6 | 76.6±0.2 | 76.6±0.9 |
| 0.15 | 83.4±0.7 | 83.9±0.6 | 83.9±0.3 | 84.5±0.5 |
| 0.25 | 85.4±0.3 | 86.0±0.4 | 86.1±0.2 | 86.2±0.3 |

and a small $\tau$ tends to reduce the swapping rates to maintain the geometric covering advantage of the batch candidates. We evaluate Shaker on CIFAR-10N (40%) with eleven different $\tau$. We run each setting three times to compute its mean value and corresponding error bar. Figure 5 visualizes the statistical impact of $\tau$. We conclude with three key observations: (1) When the budget is limited, the effect of covering radius dominates the coreset selection. For example, when the coreset size is 0.05, Eq. 5 indicates that Shaker will swap in more clean samples with a large $\tau$. However, its performance is suboptimal than the small $\tau$. (2) Low-loss samples should be prioritized over these samples with small covering radius if the budget is sufficient. For example, the test accuracy first reaches the highest point at $\tau = 0.3$ and then remains almost unchanged when the selection ratio is 0.25. (3) As the coreset budget increases, the optimal $\tau$ also increases. When the coreset size is 0.05, the optimal $\tau$ is 0.1. Then when the coreset size increases to 0.25, the optimal $\tau$ moves to 0.3. It demonstrates the importance of clean samples in a large subset.

**The Effect of Batch Size.** Batch size $B$ controls the number of samples participating in the alternating step, enabling Shaker to scale to large datasets with a small constant amount of resources. Large batch size approaches global optimal swapping results, while small size allows the alternating next step to correct the swapping errors more timely. Table 5 illustrates the impact of batch sizes $B \in \{100, 250, 500, 2500\}$ on the test accuracy of CIFAR-10N (40%) with varying coreset sizes. The performance of large batch size is 0.5%, 1.1%, and 0.8% higher than that of small batch size respectively. These results demonstrate that Shaker is weakly positively correlated with batch sizes. It suggests that a larger batch might be better within the available computing resources. In addition, a small batch can also achieve good performance.

## 6 RELATED WORK

**Coreset Selection.** From the perspective of the used information, coreset selection methods can be grouped into three categories: (1) Methods based on the geometric feature space. This branch methods assume that samples close to each other in the feature space tend to have similar properties so that they can be considered redundant and removed. $k$-Center is representative in this direction (Sener & Savarese, 2018). By optimizing the covering radius, it selects samples that can maximize the coverage of the entire dataset. Similarly, FDMat matches the feature distribution of the dataset by optimal transport (Xiao et al., 2024). In addition, some methods try to select samples near the decision boundary in the feature space as coresets (Margatina et al., 2021). (2) Methods based on the prediction outputs. They assume that the samples with low confidence during warm-up training have high training utility and should be selected into the coreset. In this direction, Margin selects samples by picking the samples with high difference between the top two highest softmax values (Coleman et al., 2020). Moderate selects the samples of moderate difficulty by evaluating their distances from the median (Xia et al., 2023b). Forgetting selects samples by counting misclassification after they have been classified correctly in warm-up training (Toneva et al., 2019). (3) Methods based on the updating model gradients. GraNd reduces datasets based on the average norm of the gradient vector of the samples (Paul et al., 2021). GradMatch seeks to make the gradient generated by the coreset as close as possible to the full dataset (Killamsetty et al., 2021a). Furthermore, some methods model coreset selection as a bilevel optimization problem (Killamsetty et al., 2021b) or a submodular optimization problem (Park et al., 2023). However, these typical coreset methods focus on one type of selection information and show limitations in complex data characteristics. For example, sample labels may be mislabeled, or sample classes could exhibit a long-tail distribution. Single-type information is not sufficient to handle such pathological data well. Our method combines the geometric feature space and the prediction outputs. We believe that coreset methods that combine two or more information will play a more important role in dealing with complex real-world data.

**Noise-Robust Training.** Plenty of works have been proposed to enhance the noise robustness of DNNs. For a comprehensive research progress, please refer to (Song et al., 2023). Among these works, we discuss three closely related branches: (1) Sample-selection-based methods (Kim et al., 2021; Patel & Sastry, 2023; Xia et al., 2023a; 2022) sound similar to coreset selection under label noise. In this regard, Jiang *et al.* select clean samples with the small-loss criterion (Han et al., 2018). Mirzasoleiman *et al.* identify clean data through the properties of the neural network Jacobian matrix (Mirzasoleiman et al., 2020). However, the primary goal of these methods is to achieve noise-robust performance, while coreset selection aims at noise-robust datasets for various downstream tasks. The unpaired goals could result in suboptimal dataset compactness and efficiency. (2) Re-labeling methods aim to correct noisy labels and incorporate them into training using heuristic strategies. DivideMix enhances this process via a co-training framework, which improves re-labeling accuracy through mutual supervision. SOP+ introduces learnable auxiliary variables and enforces a self-consistency loss to further refine label correction. (3) Noise-robust loss functions enhance the generalization performance of models by overcoming the limitations of existing loss functions. SL is a representative one. It designs noise-robust symmetric cross entropy learning with a noise-robust reverse cross entropy. The experiments in Section 5 show that Shaker performs good transferability to re-labeling and noise-robust loss functions.

Another direction is to co-design coreset selection and noise-robust learning. Prune4ReL uses the neighbor consistency of re-labeling to design coreset selection (Park et al., 2023). However, such methods are tightly coupled with downstream model training and therefore cannot be transferred to more applications. Its performance is also proven to be weaker than that of the general Shaker.

## 7 CONCLUSION

We capture the phenomenon that the performance of coresets generated with $k$-center lag in the presence of label noise. We explain this phenomenon by introducing the noise rate into the set cover theory. Empirical experimental results support our theorem. We propose a coreset selection method based on shaking $k$-center, which swaps in reliable neighbors of $k$-center while maintaining good set cover properties. Experiments on CIFAR-10N, CIFAR-100N, and large-scale WebVision show that Shaker achieves better generalization performance than baselines.

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
