# OpenReview forum: "Shake the k-Center: Toward Noise-Robust Coresets via Reliability Swapping between Neighbors"
_ICLR.cc/2026/Conference — Submitted to ICLR 2026_

### Official Review · Reviewer_iWNH · 2025-10-27

**Soundness:** 2
**Presentation:** 3
**Contribution:** 2
**Rating:** 4
**Confidence:** 4

**Summary:**

The paper studies coreset selection under label noise and explains why classic k-Center, which optimizes a feature-space covering radius, degrades on noisy labels. It decomposes the noisy-coreset selection error into three terms and derives a bound showing dependence on (i) covering radius and (ii) the coreset’s noise rate (Theorem 1) . Motivated by an empirical trade-off between these two factors, the authors propose Shaker: (1) greedily build a batch of k-Center candidates to reduce covering radius, then (2) swap each candidate with a “reliable neighbor” via a linear assignment (bipartite matching) whose cost favors close, low-loss neighbors; repeat in batches to fill the budget . Experiments on CIFAR-10N/100N and WebVision show accuracy gains over several selection baselines, with reported margins “up to 14.3%” on CIFAR-10N.

**Strengths:**

Originality.

Adds a simple but distinct twist to geometric coresets under noise: keep the k-Center “cover” but post-process by reliability-aware swapping via assignment, explicitly balancing geometry and label reliability.

Provides a compact decomposition for noisy-coreset error and a bound that surfaces coreset noise rate alongside covering radius, giving a clear lens on why vanilla k-Center fails under noise.

Quality.

The theoretical step is technically correct and transparent (triangle inequality + known k-Center bound), yielding a clean statement that guides design (minimize radius and coreset noise) .

The method is well-specified: greedy facility-location candidate selection (2-approx) and LAPJV assignment for swapping; the cost blends distance and per-example loss with a temperature τ, and an alternating scheme controls cover drift.

Clarity.

The paper cleanly moves from diagnosis (Figure 2 trade-off) to method (Figure 3) to an algorithm box; assumptions and settings are explicit, and the training/eval protocols (warm-up, optimizers, noise-robust learners) are spelled out.

Significance.

If one needs small supervised coresets under real label noise (a practical regime in pruning, NAS, continual learning), reducing noise rate and maintaining coverage is meaningful; Shaker beats several standard selectors on CIFAR-10N/100N and WebVision.

**Weaknesses:**

The problem is framed and evaluated only for image classification with class-label noise and fixed small budgets; there’s no test on other modalities, tasks (e.g., detection, NLP), or other data pathologies (class imbalance, domain shift), nor any demonstration on the motivating downstreams (e.g., NAS speed-ups or continual-learning memory) beyond accuracy tables. Consider adding a NAS-style wall-clock/accuracy trade-off or rehearsal-CL test to evidence broader impact, or at least one non-vision dataset.

Theory is largely incremental. The “new” bound is a straightforward decomposition using the triangle inequality plus the known k-Center guarantee; it clarifies factors but doesn’t yield new algorithmic guarantees for the proposed swapping (no approximation ratio or convergence/property that links Eq. (3) to the assignment procedure) . Strengthen by (i) bounding how much swapping can inflate the radius, or (ii) proving that, under assumptions, the alternating scheme reduces a surrogate for the RHS of Eq. (3).

The evaluation uses noise-robust learners (SOP+, DivideMix, SL) during downstream training, which themselves fight label noise. It’s hard to isolate how much gain comes from the coreset selection vs. the training recipe; an apples-to-apples comparison with standard CE and identical warm-up across methods would clarify attribution.

The reliability proxy is small loss after warm-up, a popular heuristic but sensitive to warm-up length and architecture. The cost in Eq. (5) (multiplicative of a logistic-loss term and exp(−distance)) and the temperature τ are somewhat ad-hoc; more principled variants (e.g., calibrated probabilities, robust losses, or uncertainty estimates) and cross-dataset tuning protocols would help.

The method repeatedly builds greedy k-Center batches and solves assignment; the paper does not give asymptotics or wall-clock for selection vs. baselines. A complexity analysis, plus timing vs. dataset size and batch size B, would strengthen practicality claims (they note future work on efficiency).

**Questions:**

1. Can you rerun a subset of experiments using plain cross-entropy (no SOP+/DivideMix/SL) to isolate Shaker’s effect on selection? Also, hold warm-up length fixed and report sensitivity across {5,10,20,40} epochs, since small-loss reliability depends on warm-up.

2. Do you track the covering radius before/after each assignment step? Can you provide a bound (or empirical curve) on radius inflation due to swapping, and perhaps a constrained assignment variant that enforces a per-swap radius cap.

3. Why the particular form in Equation (5)? Did you try alternatives (e.g. calibrated $p(y|x)$, entropy, or margin-based reliability)? Please include a small study comparing cost functions and a principled strategy to choose $\tau$ across datasets.

4. What is the end-to-end selection time vs. baselines on CIFAR-10N/100N/WebVision? Provide complexity in terms of $n$, $m$, and batch size $B$, and wall-clock on a single GPU/CPU. Any memory bottlenecks when B is large (you recommend $B=2500$)?

5. Could Shaker be extended to other dataset pathologies (class imbalance, covariate shift, long-tail)? For example, replacing “small-loss” with a reliability score that mixes per-class calibration or importance weights while keeping the assignment machinery.

6. Please add a “coverage-constrained Small-Loss” baseline: greedily ensure dispersion (e.g., k-Center on features) within each batch while selecting low-loss neighbors, to test whether assignment is essential or whether a simpler constrained greedy achieves similar gains.

---

### Official Review · Reviewer_ohf8 · 2025-10-27

**Soundness:** 2
**Presentation:** 2
**Contribution:** 2
**Rating:** 4
**Confidence:** 2

**Summary:**

This paper is motivated by the following phenomenon: The classical k-Center coreset selection approach (via covering radius in the feature space) performs poorly under label noise. This paper aims to give an explanation of this phenomenon and tries to solve this problem by proposing a coreset selection procedure that outperforms the classical k-Center coreset selection under label noise.

Concretely, this paper presents a theorem showing that the selection error of a coreset is upper-bounded by three terms: (i) loss due to label noise on the full dataset, (ii) loss due to coreset selection (proportional to the covering radius $\delta_\mathcal{S}$ as in Sener & Savarese, 2018), and (iii) loss due to noise within the selected coreset.

It then introduces the following batched selection algorithm that alternates between (1) constructing k-Center candidates with small covering radius and (2) swapping each candidate with a “reliable neighbor” (lower training loss) via a bipartite assignment objective. The effectiveness of this algorithm is evaluated by experiments.

**Strengths:**

I am from a purely theoretical background, so I may not have the expertise to fully assess the empirical evaluation. However, the proposed algorithm consistently outperforms the other selection methods listed in Tables 1, 2, and 3 across different noise rates and datasets. I interpret this as strong empirical performance, which can be a solid contribution. Taken at face value, this suggests the method is empirically strong and potentially useful in practice. (Given my limited empirical background, I note that my assessment of the experiments may not be definitive.)

**Weaknesses:**

In my view, the "theory" component of the paper is limited. The only theorem is an upper bound that follows trivially from an application of the triangle inequality to decompose the error. As such, it does not yield sharp or surprising insights.

Moreover, some claims in the paper appear stronger than what the bound supports. For example, statements such as:

> "Our theory indicates that the noise rate of the coreset constrains the generalization performance of the selected subset."


An upper bound alone does not imply necessity; to substantiate a constraint in the strong sense implied here, some form of lower bound or evidence of tightness would be needed.

While I understand that the paper contains implementations and experiments -- which may be the more significant contribution -- I feel that, given the inadequacy of the theoretical component, the abstract and introduction should be reframed to make clear that the primary contribution is empirical. In particular, I suggest removing phrases like “our complete theory demonstrates...”: the theory presented does not appear complete, as many questions remain open. For example:

-- The "seesaw effect" between covering radius and coreset noise is presented as an empirical observation that motivates the method. Given how prominently it is in the narrative of the entire paper, a theoretical analysis would significantly strengthen the paper.

-- The algorithm’s neighbor "reliability" score and the swap mechanism are heuristic. There is substantial literature (and rich theory) on swap-based local search for representative-set selection; it seems plausible to seek some theoretical guarantee (even a conditional or approximate one) for these heuristics, but no such results are provided.


-------------------------------------

As noted above, to my understanding:

-- The high-level algorithmic idea of swap-based local search around a set of k representatives is not new and has long been studied (e.g., k-medoids/PAM).
-- The specific swap metric (the measure of neighbor reliability) and the use of a linear assignment via bipartite matching might be new.

However, without explicit comparisons in the paper that highlight which components of the algorithm are novel, it is difficult to isolate the contribution and the novelty in the algorithmic design.

**Questions:**

See the above sections.

---

### Official Review · Reviewer_xwpk · 2025-10-31

**Soundness:** 3
**Presentation:** 3
**Contribution:** 3
**Rating:** 6
**Confidence:** 4

**Summary:**

This paper addresses the challenge of coreset selection for deep neural networks under label noise, focusing on the widely used k-center method. The authors provide a theoretical explanation for the degradation of k-center performance in noisy settings and introduce "Shaker," a novel algorithm that jointly optimizes covering radius and reliability by swapping in more reliable neighbors. Shaker is formalized as a bipartite graph matching problem and is shown to outperform existing baselines on noisy versions of CIFAR-10, CIFAR-100, and WebVision datasets. Extensive experiments and ablation studies demonstrate the robustness and efficiency of the proposed method, offering practical guidance for noise-robust data selection.

**Strengths:**

(1) The paper provides a theoretical framework explaining why k-center struggles under label noise, filling a gap in the literature.
(2) Shaker’s reliability swapping mechanism is innovative, combining geometric and reliability criteria through bipartite graph matching.
(3) The method consistently outperforms state-of-the-art baselines across multiple datasets and noise levels, with thorough experimental validation.
(4) The paper analyzes the impact of key hyperparameters (swapping coefficient, batch size), enhancing the credibility of the results.

**Weaknesses:**

(1) In experiments, it looks like only three coreset sizes (0.05, 0.15, 0.25) are considered. It would be better to use more coreset sizes to understand how the performance would be changed when we gradually use less and less data (e.g., from full data to a small coreset). This would help the readers better understand the effectiveness of the proposed method for different coreset sizes (e.g., small, medium, large). In addition, it looks there are some coreset methods that are specially designed to handle small coreset sizes (see [1][2] from the reference list below). It might be better to include some discussion about it (or add some additional baselines in experiments if possible).

(2) It's interesting to consider coreset problem under label noise in this paper. However, there are different types of label noise, and some papers such as [3][4] from the reference list below, also discussed methods for handling other types of label noise, e.g., one-sided label noise. It would be great if the authors can discuss the applicability of their method to other types of label noise.


**Reference**:
[1] Xia, Xiaobo, et al. "Refined coreset selection: Towards minimal coreset size under model performance constraints." arXiv preprint arXiv:2311.08675 (2023).
[2] Zheng, Haizhong, et al. "Coverage-centric coreset selection for high pruning rates." arXiv preprint arXiv:2210.15809 (2022).
[3] Liu, et al. "An analysis of boosted linear classifiers on noisy data with applications to multiple-instance learning." 2017 IEEE International Conference on Data Mining (ICDM). IEEE, 2017.
[4] Luan, et al. "Multi-Instance Learning with One Side Label Noise." ACM Transactions on Knowledge Discovery from Data 18.5 (2024): 1-24.

**Questions:**

see my comments above.

---

### Official Review · Reviewer_qfSC · 2025-11-01

**Soundness:** 1
**Presentation:** 2
**Contribution:** 1
**Rating:** 2
**Confidence:** 3

**Summary:**

This paper studies the degradation of k-center–based coreset selection under label noise and introduces Shaker, a modified k-center method designed to improve robustness. The authors first extend the classical k-center generalization bound by incorporating a label noise term and argue that the coreset quality depends jointly on the covering radius and noise rate. Based on this observation, the proposed method alternates between selecting points via k-center and replacing some selected points with nearby low-loss samples through a bipartite matching procedure. Experiments on CIFAR-10N, CIFAR-100N, and WebVision show that Shaker achieves performance improvements over baseline approaches.

**Strengths:**

- The paper addresses a practical and increasingly relevant problem—coreset selection under label noise.
- The motivation is clearly stated, and the empirical section includes comparisons across multiple noisy-label benchmarks.
- The proposed approach is conceptually simple and easy to implement, which may make it attractive for practitioners.
- The improvement on CIFAR-10N is reasonable, showing that incorporating label reliability into coreset selection can be beneficial in some settings.

**Weaknesses:**

Although the paper provides an intuitive theoretical argument by adjusting the k-center upper bound, the theoretical justification is not definitive. The decomposition primarily reiterates the original k-center analysis with an added noise term and does not establish a fundamentally new or tight characterization. As such, the theoretical contribution remains limited.

More importantly, the proposed swapping mechanism lacks a rigorous guarantee. There is no proof that the method either reduces the effective noise rate or preserves the covering property. As a result, the approach functions largely as a heuristic, and the absence of a formal guarantee weakens the claimed contribution.

Regarding the experiments, while Shaker achieves notable gains on CIFAR-10N, the improvements on the other datasets (CIFAR-100N and WebVision) are relatively small. Given the heuristic nature of the method, these modest gains raise questions about the broader impact and general applicability of the approach. Without stronger theoretical support, the limited improvements beyond CIFAR-10N make the overall contribution less convincing.

**Questions:**

- Can the authors provide a more formal justification that the proposed noise-augmented generalization bound offers new insight beyond the classical k-center analysis?
- Is there any guarantee that the swapping step maintains or improves coverage quality? If not, how should we interpret the resulting coreset from a theoretical standpoint?
- The empirical gains on CIFAR-100N and WebVision are quite limited. Are there specific scenarios where Shaker is expected to produce substantial improvements beyond CIFAR-10N?

---

### Official Review · Reviewer_ALXq · 2025-11-05

**Soundness:** 2
**Presentation:** 2
**Contribution:** 2
**Rating:** 6
**Confidence:** 3

**Summary:**

The paper proposes an explanation for the degradation in performance of k-center based coresets in presence of noise. It proposes an improved k-center based coreset algorithm in presence of noise by first creating a k-center based coreset and then swapping some of the centers to make the coreset more robust. The efficiency of the algorithm is validated by extensive experimentation on real world datasets.

**Strengths:**

1) There is relatively less work in the domain of coresets in noisy settings. Also, k-center based coresets are very intuitive. As such the ideas in this paper will be of interest to the community.
2) Overall, the paper is well written and not difficult to follow. The claims appear sound to the best of my understanding.
3) Good empirical evaluations to validate the utility of the algorithm.

**Weaknesses:**

My main concern with this paper is in terms of novelty. The paper seems to borrow a lot of ideas from the Sener and Savarese (2018) paper. The explanation of the coreset performance degradation is very similar to the ideas in that paper. Also, the 2018 paper also has a robust $k$-center algorithm. Why is that algorithm not 'robust' enough? Also swapping of center to improve a solution is also a well -known idea in local search techniques-based approximation algorithms.  I believe the authors need to explain the technical challenges and their contribution better to highlight the novelty of the paper in terms of both algorithm and proof techniques for a venue like ICLR.

**Questions:**

See Weaknesses.

---

### Meta-Review · Area_Chair_WYKm · 2026-01-04

**Summary:**

1. The theoretical analysis for the proposed algorithm is not solid enough. It is more like a heuristic algorithm.
2. The technical novelty is unclear, compared to existing coreset methods, e.g., the Sener and Savarese (2018) paper.
3. The experimental improvement compared to baselines is not significant.

**Reviewer Concerns:**

There is no rebuttal.

**Reviewer Scores:**

No one will change the score.

---

### Decision · Program_Chairs · 2026-01-26

Reject